# Fasting Proinsulin Independently Predicts Incident Type 2 Diabetes in the General Population

**DOI:** 10.3390/jpm12071131

**Published:** 2022-07-12

**Authors:** Sara Sokooti, Wendy A. Dam, Tamas Szili-Torok, Jolein Gloerich, Alain J. van Gool, Adrian Post, Martin H. de Borst, Ron T. Gansevoort, Hiddo J. L. Heerspink, Robin P. F. Dullaart, Stephan J. L. Bakker

**Affiliations:** 1Department of Internal Medicine, Division of Nephrology, University Medical Center Groningen, University of Groningen, 9713 GZ Groningen, The Netherlands; w.a.dam01@umcg.nl (W.A.D.); t.szili-torok@umcg.nl (T.S.-T.); a.post01@umcg.nl (A.P.); m.h.de.borst@umcg.nl (M.H.d.B.); r.t.gansevoort@umcg.nl (R.T.G.); s.j.l.bakker@umcg.nl (S.J.L.B.); 2Translational Metabolic Laboratory, Department of Laboratory Medicine, Radboud Institute for Molecular Life Sciences, Radboud University Medical Center, 6525 GA Nijmegen, The Netherlands; jolein.gloerich@radboudumc.nl (J.G.); alain.vangool@radboudumc.nl (A.J.v.G.); 3Department of Clinical Pharmacy and Pharmacology, University Medical Center Groningen, University of Groningen, 9713 GZ Groningen, The Netherlands; h.j.lambers.heerspink@umcg.nl; 4Department of Internal Medicine, Division of Endocrinology, University Medical Center Groningen, University of Groningen, 9713 GZ Groningen, The Netherlands; dull.fam@12move.nl

**Keywords:** type 2 diabetes, proinsulin, kidney dysfunction, hypertension, predictive value

## Abstract

Fasting proinsulin levels may serve as a marker of β-cell dysfunction and predict type 2 diabetes (T2D) development. Kidneys have been found to be a major site for the degradation of proinsulin. We aimed to evaluate the predictive value of proinsulin for the risk of incident T2D added to a base model of clinical predictors and examined potential effect modification by variables related to kidney function. Proinsulin was measured in plasma with U-PLEX platform using ELISA immunoassay. We included 5001 participants without T2D at baseline and during a median follow up of 7.2 years; 271 participants developed T2D. Higher levels of proinsulin were associated with increased risk of T2D independent of glucose, insulin, C-peptide, and other clinical factors (hazard ratio (HR): 1.28; per 1 SD increase 95% confidence interval (CI): 1.08–1.52). Harrell’s C-index for the Framingham offspring risk score was improved with the addition of proinsulin (*p* = 0.019). Furthermore, we found effect modification by hypertension (*p* = 0.019), eGFR (*p* = 0.020) and urinary albumin excretion (*p* = 0.034), consistent with an association only present in participants with hypertension or kidney dysfunction. Higher fasting proinsulin level is an independent predictor of incident T2D in the general population, particularly in participants with hypertension or kidney dysfunction.

## 1. Introduction

Insulin resistance and β-cell dysfunction are well-established, pathophysiological mechanisms underlying the development of type 2 diabetes (T2D) [1,2]. In the past decades, plasma insulin has been established as a clinical biomarker of insulin resistance [3]. Insulin and C-peptide are processed from a premature, 86-amino-acid-long peptide called proinsulin in β-cells [4,5]. Proinsulin is synthesized in the endoplasmic reticulum and is cleared from the plasma at a slower rate compared to insulin [6]. Since high levels of glucose stimulate proinsulin synthesis and secretion, elevated concentrations of proinsulin are observed in glucose-intolerant participants [7].

Both fasting proinsulin and C-peptide levels have been found to be strongly associated with insulin resistance and incident T2D [8,9]. Furthermore, previous studies showed that the kidney contributes substantially to the removal of insulin, C-peptide, and proinsulin, mainly through their degradation rather than by urinary excretion, which indicates that the kidney represents a major site for insulin metabolism and is the main organ responsible for the degradation of proinsulin and C-peptide [10,11]. Notably, we previously demonstrated that C-peptide is not a reliable marker for predicting T2D in people with hypertension and kidney dysfunction [12].

The predictive value of proinsulin for the risk of developing T2D compared to insulin and C-peptide is currently unknown, and, to the best of our knowledge, there are no longitudinal studies investigating proinsulin level as a marker of incident T2D in participants with hypertension or kidney dysfunction. Therefore, we aimed (1) to examine the association between proinsulin levels and risk of incident T2D independent of plasma insulin and C-peptide, (2) to evaluate the predictive value of proinsulin compared to insulin and C-peptide for the risk of incident T2D added to a base model of clinical predictors, and (3) to examine potential effect modification by variables related to kidney function.

## 2. Materials and Methods

### 2.1. Design and Study Population

The study was performed within the frame of the Prevention of Renal and Vascular End-Stage Disease (PREVEND) study, an observational, general-population-based, longitudinal cohort study, which investigated vascular and renal damage among inhabitants of the city of Groningen, The Netherlands, as reported in detail elsewhere [13]. Briefly, all residents of Groningen aged 28 to 75 years were invited to participate in this study from 1997 to 1998. Pregnant women and participants with type 1 diabetes and T2D using insulin were excluded. After further exclusion of individuals unable or unwilling to participate in the study, a total of 6000 individuals with a urinary albumin concentration of 10 mg/L or greater and a randomly chosen control group of 2592 individuals with a urinary albumin concentration of less than 10 mg/L completed the screening protocol and constituted the PREVEND cohort (*n* = 8592). A second screening was performed from 2001 to 2003 with 6894 participants, which was the baseline of the present study. For the current study, we excluded participants with diabetes at baseline or missing data on diabetes status or glucose levels at baseline, those with missing proinsulin values or covariate data at baseline, and participants with no follow-up data on the incidence of T2D, leaving 5001 participants for the current analyses (Appendix A).

The protocol for the PREVEND study was approved by the local ethics committee of the University Medical Center Groningen (approval number: MEC96/01/022). All participants included in the present analysis provided written, informed consent to participate, and the study procedures were conducted according to the Declaration of Helsinki.

### 2.2. Data Collection

Each screening comprised two visits to an outpatient clinic separated by three weeks. Prior to the first visit, each participant was asked to fill out a questionnaire regarding demographic data, cardiovascular and renal disease, family history of diabetes, and information about use of medication that was confirmed by using data from the pharmacy registries of all community pharmacies in the city of Groningen [14]. Smoking and alcohol use were based on self-reports. Smoking status was categorized as never, present, or former. Self-reported alcohol consumption was assessed using a question in which participants were asked to choose 1 of the following categories: abstention (no alcohol consumption), 1 to 4 units/month, 2 to 7 units/week, 1 to 3 units/day, or >3 units/day [15]. Height and weight were measured with the participants standing without shoes and heavy outer garments. Body mass index (BMI) was calculated by dividing weight in kilograms by height in meters, squared. Both in the first and second visit, systolic and diastolic blood pressure were recorded in the supine position with an automatic Dinamap XL Model 9300 series device (Johnson & Johnson, New Brunswick, NJ, USA). Hypertension was defined by self-reported physician diagnosis, use of antihypertensive medication, or blood pressure ≥ 140/90 mmHg. Two 24 h urine specimens were collected in the last week before the second visit. At the second screening visit, venous blood samples were drawn from an antecubital vein after a 10 h fast, and EDTA plasma samples were stored at −80 °C.

### 2.3. Laboratory Measurements

High-density lipoprotein (HDL) cholesterol and triglycerides concentration were measured on a Beckman Coulter AU Analyzer (Randox Laboratories, Crumlin, UK). Total cholesterol and fasting plasma glucose (FPG) were measured by dry chemistry (Eastman Kodak, Rochester, NY, USA). Urinary albumin excretion (UAE), given as the mean of the two 24 h urine albumin excretions, was quantified using a nephelometry with a threshold of 2.3 mg/L (Dade Behring Diagnostic, Marburg, Germany). Serum creatinine and cystatin C measurement were carried out as described previously [16]. Estimated glomerular filtration rate (eGFR) was calculated by using the Chronic Kidney Disease Epidemiology Collaboration (CKD-EPI) equation, which combines information on serum creatinine and serum cystatin C [17]. Fasting insulin was measured with an AxSYM autoanalyzer (Abbott Diagnostics, Amstelveen, The Netherlands). Homeostasis model assessment for insulin resistance (HOMA-IR) was calculated as fasting plasma insulin (µU/mL) × fasting plasma glucose (mmol/L)/22.5 [18]. C-peptide was measured in plasma with an electrochemiluminescent immunoassay using a Cobas e602 (Roche Modular E, Roche Diagnostics, Mannheim, Germany). Proinsulin was measured in plasma with U-PLEX platform using ELISA (Metabolic Combo 1, K15281K, Meso Scale Discovery, Rockville, MD, USA). Two monoclonal antibodies were directed against separate antigenic determinants on the proinsulin molecule. The analytical measurement range was 0.03 to 126 pmol/L.

### 2.4. Outcome Ascertainment

Follow-up time was defined as the period between the second screening round (baseline) and the date of ascertainment of T2D. Incident T2D was ascertained if one or more of the following criteria were met: (1) fasting plasma glucose (FPG) ≥ 7.0 mmol/L (126 mg/dL); (2) random sample plasma glucose ≥ 11.1 mmol/L (200 mg/dL); (3) self-report of a physician diagnosis of T2D; and (4) initiation of glucose-lowering medication use, retrieved from a central pharmacy registry [14].

### 2.5. Statistical Analyses

All analyses were conducted using the statistical packages IBM SPSS (version 24.0.1; SPSS, Chicago, IL, USA) and STATA/SE (version 14; StataCorp, College Station, TX, USA). For all analyses, 2-sided *p*-values < 0.05 were considered statistically significant. The study population characteristics were analyzed separately for quintiles of proinsulin. For analyses, we combined quintiles 1–3 as one group and compared this group with quintile 4 and quintile 5. Continuous data were expressed as mean ± standard deviation (SD) or as median and interquartile range (IQR) for normal distribution and skewed distribution of continuous values, respectively. Categorical data were evaluated by means of a chi-squared test. Differences among the 3 quintile groups (see above) were determined by using *p*-value for trends or a Kruskal–Wallis test when applicable. Uni- and multivariable linear regression were used to determine the presence of an association between participants’ characteristics and proinsulin levels. Variables with a skewed distribution were loge transformed. Incident T2D was visualized by Kaplan–Meier curve according to quintile 5, quintile 4, and the three lowest quintiles of proinsulin levels, with statistical significance tested by log-rank test.

We applied Cox proportional hazards regression analysis to study the prospective association of proinsulin level with the risk of incident T2D. Hazards ratios (HR) with 95% confidence intervals (CIs) were calculated per 1 SD increment of proinsulin (loge transformed). Additionally, these associations were evaluated across the quintiles of proinsulin where the combined quintiles 1–3 were assigned the reference category. First, we calculated HRs (95% CIs) for the crude model. In model 1, HRs (95% CIs) were calculated after adjustment for age and sex. In model 2, we further adjusted for smoking status and alcohol consumption. In model 3, we further adjusted for BMI, family history of diabetes, and hypertension. In model 4, we calculated HRs (95% CIs) after further adjustment for triglycerides, total cholesterol, and HDL-C, which may be confounders of the association between proinsulin and risk of T2D. In model 5, we further adjusted for UAE and eGFR. In models 6 and 7, we further adjusted for FPG and insulin, respectively. In the last model, we calculated HRs (95% CIs), which were further adjusted for C-peptide. The proportional hazards assumption for the models was tested to confirm absence of violation.

We examined the risk of developing T2D with and without inclusion of proinsulin for the Framingham offspring (FOS) risk score by testing for differences in Harrell’s C-statistics [19]. Additionally, to assess the added value of proinsulin, improvement of T2D prediction was examined in terms of discrimination and integrated discrimination improvement (IDI). The IDI parameter was calculated by subtracting the mean difference of predicted risk between the FOS risk score and the FOS risk score including proinsulin [20].

For secondary analyses, we tested interactions to assess statistical evidence for effect modification by sex, age, BMI, eGFR, UAE, glucose, insulin, and hypertension in the multivariable analyses. For predictors of which interaction test was significant, we performed subgroup analyses in HRs across categories of prespecified subject characteristics, including eGFR (<90 mL/min/1.73 m^2^ vs. ≥90 mL/min/1.73 m^2^), UAE (<15 mg/24 h vs. ≥15 mg/24 h), and hypertension (no vs. yes) in the last model (model 8).

In sensitivity analyses, we applied competing risk analyses based on Fine and Gray’s proportional subhazards model since a death event that happened prior to the T2D event could prevent the individuals from experiencing T2D development.

## 3. Results

### 3.1. Baseline Characteristics

Baseline median (IQR) proinsulin in the 5001 selected individuals was 6.89 (5.15–9.55) pmol/L. Baseline characteristics of the participants according to the three groups of proinsulin levels (combined lowest three quintiles, 1–3; quintile 4; and quintile 5, separately) are shown in Table 1. Participants with higher proinsulin levels were more likely to be men, older, to consume less alcohol, and to have smoked more frequently in the past but smoked less frequently at baseline. These same participants had a higher likelihood of a family history of diabetes and past history of gestational diabetes. They also had higher BMI, blood pressure, total cholesterol, triglycerides, glucose, insulin, C-peptide, and HOMA-IR but lower HDL-C. They used antihypertensive medications and lipid-lowering drugs more frequently. Additionally, participants with higher proinsulin levels had higher serum creatinine and higher UAE but lower eGFR.

### 3.2. Cross-Sectional Associations

The correlations between loge-transformed proinsulin levels and variables of interest due to their association with T2D are shown in Appendix A. In univariable regression analyses, age, parental history of diabetes, BMI, systolic blood pressure, total cholesterol, triglycerides, use of antihypertensive and lipid-lowering medication, FPG, insulin, C-peptide, and UAE were positively associated with proinsulin, whereas female sex, HDL-C, and eGFR were inversely associated with proinsulin. In the multivariable model, which included all the variables, significant, independent positive associations with proinsulin were observed for age, BMI, triglycerides, FPG, insulin, C-peptide, and UAE, whereas inverse independent associations were observed for female sex and total cholesterol.

### 3.3. Association of Proinsulin with Risk of Developing T2D

During a median follow-up period of 7.2 years (IQR, 6.0–7.7 years), 271 participants developed T2D. Higher proinsulin levels were associated with a higher risk of incident type 2 diabetes compared to lower proinsulin levels (*p* < 0.001) (Figure 1). Subsequently, we proceeded with longitudinal analyses for quintiles of proinsulin and incident type 2 diabetes (Table 2). Higher levels of proinsulin were associated with an increased risk of type 2 diabetes in crude analyses and after adjustment for age and sex in model 1. The association remained significant after further adjustment for other variables. In model 2, we further adjusted for smoking status and alcohol consumption. In model 3, we further adjusted for BMI, family history of diabetes, and hypertension. In model 4, we further adjusted for triglycerides, total cholesterol, and HDL-C. In model 5, we further adjusted for UAE and eGFR. In models 6 and 7, we further adjusted for FPG and insulin, respectively. Further adjustment for C-peptide attenuated the association of proinsulin with incident T2D, but the associations remained significant (model 8). Comparable results were obtained when proinsulin level was examined per 1SD change (Table 2).

### 3.4. Model Performance Compared with the FOS Risk Score

The longitudinal association of proinsulin with incident T2D was independent of age, sex, BMI, family history of diabetes, blood pressure, triglycerides, HDL cholesterol, and FPG, all components of the FOS risk score [19]. Harrell’s C-index for the FOS risk score amounted to 0.886, which was significantly improved to 0.888 with the addition of proinsulin and to 0.888 with the addition of C-peptide (Table 3). However, addition of insulin to the FOS risk score did not improve Harrell’s C-index. The IDI was significantly changed after addition of proinsulin and C-peptide to the FOS risk score to 0.0069 (*p* = 0.004) and 0.0056 (*p* = 0.02), respectively. The IDI was not significantly changed after addition of insulin to the FOS risk score (Table 3).

### 3.5. Secondary Analyses of Proinsulin with Incident T2D in Various Groups

To find potential effect modifications, we tested for interactions by sex, age, BMI, eGFR, UAE, FPG, insulin, and hypertension. In multivariable analyses, we found significant effect modification for eGFR (*p* = 0.019), UAE (*p* = 0.030) and hypertension (*p* = 0.019). Consequently, secondary analyses were performed among subgroups of participants with high eGFR (≥90 mL/min/1.73 m^2^), low eGFR (<90 mL/min/1.73 m^2^), UAE < 15 mg/24 h, UAE ≥ 15 mg/24 h, hypertension, and without hypertension in the last model (Figure 2). The association between proinsulin levels and incident T2D was significant in participants with kidney dysfunction (eGFR < 90 mL/min/1.73 m^2^ or UAE ≥ 15 mg/24 h) or hypertension after adjustment for C-peptide (Figure 2).

### 3.6. Sensitivity Analyses on Proinsulin and T2D

The competing risk of T2D and death may confound the estimated risk for T2D as an outcome, which means a death event that happened prior to the T2D event could prevent individuals from experiencing T2D development. Therefore, for the sensitivity analysis, we evaluated the association between proinsulin and incident T2D where we restricted the outcome to incident type 2 diabetes and censored for death. Higher proinsulin was significantly associated with increased risk of incident T2D in the crude analyses. This finding remained materially unchanged in further multivariable-adjusted analyses after further adjustments for all confounders similar to the primary analyses, which were first adjusted for age and sex in model 1. Subsequently, we further adjusted for smoking status and alcohol consumption in model 2, and additionally adjusted for BMI, family history of diabetes, and hypertension in model 3. We further adjusted for triglycerides, total cholesterol, and HDL cholesterol in model 4, and additionally adjusted for eGFR and urinary albumin excretion in model 5. Then, we further adjusted for glucose, insulin, and C-peptide in models 6, 7, and 8 (Appendix A).

## 4. Discussion

In this large cohort of a predominantly middle-aged population, we found fasting proinsulin levels were independently associated with incident T2D. The association remained statistically significant after multivariable adjustment for BMI, hypertension, glucose, and relevant lipid-related biomarkers, as well as after adjustment for plasma insulin and C-peptide. The addition of proinsulin to the FOS risk model improved the prediction value of T2D. Moreover, the association of proinsulin with incident T2D was particularly strong in participants with hypertension or kidney dysfunction.

Proinsulin levels have been shown to be associated with T2D parameters, including glucose and insulin resistance, in different cohorts [21,22,23,24,25]. Similarly, in the cross-sectional analyses, we found strong correlations between proinsulin and risk factors of T2D such as BMI, systolic blood pressure, HDL cholesterol, triglycerides, and FPG. Moreover, during the follow up, we found significant association between proinsulin and incident T2D, which was inconsistent with the initial cross-sectional analyses. Two large studies in men and women in Finland and the US, respectively, reported that higher proinsulin levels are associated with an increased risk of developing T2D [24,26].

We used baseline fasting proinsulin levels to investigate development of T2D. At the early stage of T2D progression, insulin resistance in peripheral tissues increases insulin demands and finally leads to an impairment of β-cell function in later disease stages [2,27]. Elevation of fasting proinsulin, as a clinical biomarker of insulin resistance, is very closely correlated to insulin resistance, assessed by intravenous (IV) glucose tolerance test and by HOMA-IR [8,28,29]. Diabetes development may lead to increased proinsulin release, which finally induces a deterioration of β-cell function [8]. Accordingly, proinsulin is known as a marker of β-cell dysfunction [30]. Moreover, β-cell loss, as well as α-cell hyperplasia, is found in chronic hyperglycemia. Impaired α-cell function has been recently introduced as a pathophysiological factor for diabetes development which has been linked to dysregulation of glucagon secretion [31]. While glucagon is primarily released from α-cells, which enhance insulin secretion, the role of glucagon and α-cells and their associations with proinsulin levels in diabetes progression need to be evaluated in future studies. In the current study, we measured fasting proinsulin by using a specific ELISA immunoassay which was recently developed. This assay could distinguish between intact proinsulin and its specific and unspecific cleavage products without cross-activity [32,33]. The use of these assays in recent epidemiological and intervention studies implies that proinsulin level can be implemented as a feasible marker that could be used in routine laboratories.

Our findings showed strong and independent association between proinsulin and incident T2D. This association was observed according to independent association after adjustment for other strong predictors of incident T2D. In comparison with previous studies, the association remained significant after adjustment for potential confounders and biomarkers that are traditionally viewed as strong predictors of incident T2D such as fasting C-peptide and fasting insulin. C-peptide was demonstrated as another useful marker of T2D development in large population studies: better than insulin [12,34]. We previously confirmed that fasting C-peptide level could predict T2D by significantly improving the FOS risk score prediction model if added to the FOS risk score [12,19]. In the current study, evaluation of the prognostic value of proinsulin showed a significant improvement in the FOS risk score for the prediction of diabetes when proinsulin was added to the FOS risk score. These results provide new information that proinsulin is a reliable biomarker for predicting incident T2D beyond other established clinical markers including age, sex, BMI, family history of diabetes, blood pressure, triglycerides, HDL cholesterol, and FPG [19].

We found that elevated proinsulin is associated with increased risk of incident T2D in participants with hypertension or kidney dysfunction. Although the association between proinsulin and incident T2D has been investigated in men with metabolic syndrome, potential effect modification by hypertension or kidney dysfunction was not investigated [26]. An association between proinsulin and hypertension was shown in previous studies, with stimulation of renal sodium retention as the potential underlying mechanisms [35,36]. Moreover, insulin, C-peptide, and proinsulin are removed by kidneys and, therefore, renal dysfunction may affect the renal extraction of those peptides, but this is much less applicable to proinsulin and insulin than to C-peptide, because the rate of renal removal of C-peptide is seven times higher than that of insulin and even 50 times higher than that of proinsulin [10]. This may indicate the kidneys as a major site for C-peptide removal and much less so for proinsulin. Moreover, hepatic extraction of proinsulin is also very low, which may explain its very long half-life in the circulation and high plasma concentration in the fasting state [37]. In the past, we found a strong association between proinsulin and incident post-transplantation diabetes in renal transplant recipients which is suggestive of a particular ability of proinsulin to serve as a marker of β-cell dysfunction in subjects with kidney dysfunction [38]. In line with this, we found that proinsulin was associated with incident T2D in participants with kidney dysfunction (eGFR < 90 mL/min/1.73 m^2^ or urinary albumin excretion ≥ 15 mg/24 h). These findings raise the possibility that proinsulin as a clinical biomarker can be helpful for recognizing individuals at higher risk of T2D, especially people with other comorbidities, including hypertension and kidney dysfunction.

The association between proinsulin and all-cause and cardiovascular mortality was investigated in previous studies [39,40]. It was suggested that this was due to the attribution of proinsulin to β-cell dysfunction and insulin resistance. The increased risk of cardiovascular or overall mortality was found in individuals with higher proinsulin levels, independent of insulin resistance and glucose tolerance status [40]. Those studies showed that proinsulin may predict mortality. Accordingly, the result from competing risk analyses with mortality in our population showed that the association between proinsulin and T2D remained similar to the primary results, suggesting that higher levels of proinsulin are associated with incident T2D even when death events were excluded.

The strength of the current study is that it included a large number of participants with a varied age range and a follow-up period of 7.3 years. Furthermore, in secondary analyses, we were able to perform subgroup analyses in participants with hypertension or kidney dysfunction. Our study’s limitation was that the majority of the participants in our study were Caucasian, precluding extrapolation to other ethnicities. Nonetheless, our results are in line with previous studies which were performed in a population with different ethnicities. Proinsulin was significantly elevated in Korean patients with T2D and glucose intolerance [6]. In a multicenter study, which consisted of non-Hispanic white and black individuals, increased proinsulin levels predicted incident T2D [23]. Similar results about the positive association between proinsulin levels and incident T2D were found in white American and Finnish populations as well [24,26].

## 5. Conclusions

In conclusion, a higher fasting proinsulin level, measured by new ELISA assays, is associated with an increased risk of incident T2D independent of glucose, insulin, C-peptide, and other clinical factors in the general population. The association was particularly significant in participants with hypertension or kidney dysfunction.

## Figures and Tables

**Figure 1 jpm-12-01131-f001:**
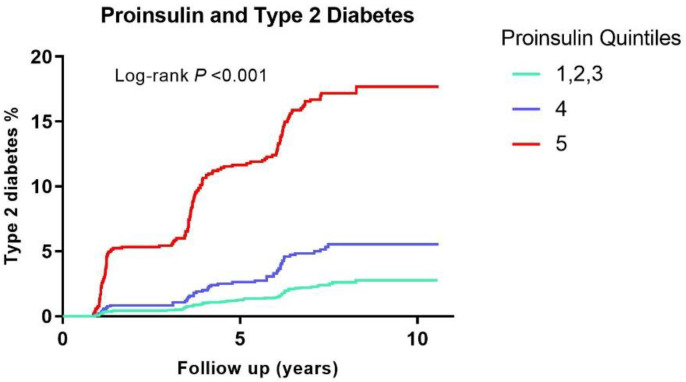
Kaplan–Meier curves for diabetes survival according to quintiles of proinsulin in 5001 participants of the PREVEND study. PREVEND: Prevention of Renal and Vascular End-Stage Disease.

**Figure 2 jpm-12-01131-f002:**
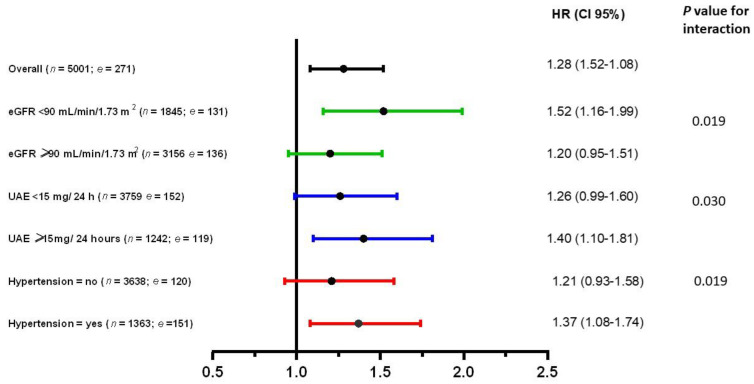
Association between proinsulin and risk of diabetes in the overall population and stratified by selected characteristics (*n* = number of participants, *e* = number of events). Multivariable hazard ratios (95% confidence intervals) for risk of incident T2D are expressed per log unit increase proinsulin levels. Hazard ratios (95 CIs) were derived from Cox proportional hazards regression models adjusted for age, sex, smoking status, alcohol consumption, BMI, family history of diabetes, triglycerides, total cholesterol, HDL cholesterol, eGFR, urinary albumin excretion, glucose, and insulin and C-peptide. BMI: body mass index; HDL: high-density lipoprotein; eGFR: estimated glomerular filtration rate.

**Table 1 jpm-12-01131-t001:** Baseline characteristics according to quintiles of proinsulin in 5001 participants of the PREVEND study.

	Quintiles	1,2,3	4	5	*p*-Value for Trend *
Proinsulin (pmol/L)		<7.78	7.78–10.37	>10.37	
Participants, N		3001	1000	1000	
Female (%)		58.0	43.6	35.5	<0.001
Age		49.9 ± 10.8	54.4 ± 11.5	57.6 ± 11.3	<0.001
Race, white (%)		99.2	99.0	99.6	0.329
Family history of diabetes (%)		15.5	18.3	22.1	<0.001
Smoking status,					0.004
Never (%)		31.0	29.5	26.2	
Current (%)		29.4	25.0	23.6	
Former (%)		39.0	45.5	50.0	
Alcohol consumption,					<0.001
None (%)		21.7	23.8	28.4	
1–4 units per month (%)		16.5	17.9	17.5	
2–7 units per week (%)		34.6	30.9	28.9	
1–3 units per day (%)		23.3	22.7	21.2	
>3 units per day (%)		4.0	4.6	3.9	
Gestational diabetes (%)		1.0	2.3	2.8	0.002
Length (cm)		172 ± 8	173 ± 9	173 ± 8	0.002
Weight (kg)		75.4 ± 12.4	81.4 ± 13.0	88.9 ± 15.3	<0.001
BMI (kg/m^2^)		25.2 ± 3.6	27.1 ± 3.8	29.4 ± 4.5	<0.001
Systolic blood pressure (mmHg)		121.0 ± 16.5	128.1 ± 17.7	133.5 ± 19.0	<0.001
Diastolic blood pressure (mmHg)		71.4 ± 8.7	74.7 ± 8.8	76.5 ± 8.5	<0.001
Use of antihypertension medication (%)		9.6	18.3	28.2	<0.001
Hypertension (%)		18.9	33.1	46.4	<0.001
Cholesterol (mmol/L)		5.3 ± 1.0	5.6 ± 0.9	5.5 ± 1	<0.001
HDL (mmol/L)		1.3 ± 0.3	1.2 ± 0.3	1.1 ± 10.3	<0.001
Triglycerides (mmol/L)		0.9 (0.7–1.3)	1.2 (0.8–1.7)	1.5 (1.1–2.1)	<0.001
Use of lipid-lowering medication (%)		4.9	9.2	15.3	<0.001
Glucose (mmol/L)		4.7 ± 0.5	4.9 ± 0.6	5.2 ± 0.7	<0.001
Insulin (mU/L)		6.7 (5.0–9.1)	9.1 (6.7–12.2)	13.2 (9.2–19.3)	<0.001
C-peptide (pmol/L)		624 (514–771)	789 (654–984)	1078 (861–1353)	<0.001
HOMA-IR ((mU mmol/L)/22.5)		1.4 (1.0–1.9)	1.9 (1.4–2.7)	3.0 (2.0–4.5)	<0.001
Serum creatinine (μmol/L)		70.2 ± 12.8	73.6 ± 14.0	78.5 ± 30.7	<0.001
Plasma albumin (g/L)		43.9 ± 5.5	43.8 ± 2.6	43.9 ± 2.6	0.22
eGFR (mL/min/1.73 m^2^)		97.5 (86.9–106.9)	94.1 (81.6–104.4)	87.7 (75.6–98.9)	<0.001
UAE (mg/24 h)		7.7 (5.8–11.8)	9.4 (6.3–17.3)	11.7 (7.1–25.4)	<0.001

Continuous variables are reported as mean ± SD or median (interquartile range), and categorical variables are reported as percentage. * Determined by linear-by-linear association chi-square test (categorical variables) and linear regression (continuous variables). BMI: body mass index; HDL: high-density lipoprotein; HOMA-IR: homeostatic model assessment of ß-cell function and insulin resistance; eGFR: estimated glomerular filtration rate; UAE: urinary albumin excretion; PREVEND: Prevention of Renal and Vascular End-Stage Disease.

**Table 2 jpm-12-01131-t002:** Association between proinsulin and risk of incident type 2 diabetes (T2D) in 5001 participants of the PREVEND study.

	Quintiles of Plasma Proinsulin, pmol/L				Proinsulin Per 1 SD Increase	
						*p*-Value
		1,2,3	4	5		
Cases		66	49	156	271	
Crude analysis		1.00 (ref)	2.24 (1.55–3.25)	8.10 (6.07–10.81)	2.66 (2.41–2.93)	<0.001
Model 1		1.00 (ref)	1.99 (1.37–2.90)	6.69 (4.94–9.06)	2.55 (2.29–2.84)	<0.001
Model 2		1.00 (ref)	1.99 (1.37–2.90)	6.62 (4.89–8.97)	2.53 (2.27–2.81)	<0.001
Model 3		1.00 (ref)	1.56 (1.07–2.27)	3.84 (2.79–5.29)	2.05 (1.81–2.32)	<0.001
Model 4		1.00 (ref)	1.44 (0.98–2.09)	3.14 (2.27–4.35)	1.88 (1.65–2.15)	<0.001
Model 5		1.00 (ref)	1.41 (0.96–2.08)	3.18 (2.28–4.44)	1.90 (1.66–2.18)	<0.001
Model 6		1.00 (ref)	1.08 (0.73–1.60)	1.90 (1.34–2.68)	1.37 (1.18–1.59)	<0.001
Model 7		1.00 (ref)	1.06 (0.71–1.57)	1.84 (1.27–2.65)	1.37 (1.16–1.61)	<0.001
Model 8		1.00 (ref)	1.05 (0.70–1.56)	1.51 (1.02–2.23)	1.28 (1.08–1.52)	0.005

HRs (95% CIs) were derived from design-based Cox proportional hazard models. Proinsulin is loge transformed. Model 1 is adjusted for age and sex. Model 2 is additionally adjusted for smoking status and alcohol consumption. Model 3 is additionally adjusted for BMI, family history of diabetes, and hypertension. Model 4 is additionally adjusted for triglycerides, total cholesterol, and HDL cholesterol. Model 5 is additionally adjusted for eGFR and urinary albumin excretion. Model 6 is additionally adjusted for glucose. Model 7 is additionally adjusted for insulin. Model 8 is additionally adjusted for C-peptide.

**Table 3 jpm-12-01131-t003:** Additive value of proinsulin, C-peptide, and insulin for the prediction of T2D in 5001 participants of the PREVEND cohort.

	C-Statistics	*p*-Value for Change in C-Statistics	IDI	*p*-Value
FOS risk model	0.886 (0.867–0.906)	-	-	-
+ Proinsulin	**0.888 (0.869–0.907)**	**0.019**	**0.0069**	**0.004**
+ C-peptide	**0.888 (0.870–0.908)**	**0.018**	**0.0056**	**0.02**
+ Insulin	0.887 (0.867–0.906)	0.241	0.0020	0.112

FOS risk score, including age, sex, BMI, family history of diabetes, systolic blood pressure, diastolic blood pressure, triglycerides, total cholesterol, HDL cholesterol, and FPG. Statistically significant variables are given in bold print. FOS: Framingham offspring risk score; NRI: net reclassification index; IDI: integrated discrimination improvement; HDL: high-density lipoproteins; BMI: body mass index; FPG: fasting plasma glucose.

## Data Availability

The data presented in this study are available on request from the corresponding author.

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
