# Peer review of "Fasting Proinsulin Independently Predicts Incident Type 2 Diabetes in the General Population"

_jpm, 2022, doi:10.3390/jpm12071131_

Round 1
Reviewer 1 Report
The manuscript by Sokooti and colleagues, is to my knowledge, one of the first to conclusively describe the predictive value of fasting proinsulin for incident T2DM. The study is well written, with a very detailed methods section and clearly presented data. The authors should be commended on the completion of such a large scale longitudinal study. I have only minor comments:
1. In the text associated with Table 2, it would be useful to be explicit that the models were additionally adjusted for each of the variables. I note that this is done well in the legend to the table, but wasn't so evident from the text.
2. the description of sensitivity analysis (section 3.6) is rather brief and would benefit from further detail including why the outcomes were limited and what the additional multivariate analysis included.
3. the authors acknowledge that generalising their data to other ethnicities is not possible given the predominantly white (99%) population studied. They do however, make reference to other studies in other ethnicities. If would be very useful for the reader to have a short description of the main findings of these studies.
Author Response
Response to Reviewer 1 comments
The manuscript by Sokooti and colleagues, is to my knowledge, one of the first to conclusively describe the predictive value of fasting proinsulin for incident T2DM. The study is well written, with a very detailed methods section and clearly presented data. The authors should be commended on the completion of such a large scale longitudinal study. I have only minor comments:
Response. We would like to thank the Reviewer for the thoughtful insights and remarks.
- In the text associated with Table 2, it would be useful to be explicit that the models were additionally adjusted for each of the variables. I note that this is done well in the legend to the table, but wasn't so evident from the text.
Response 1. We thank the reviewer for this remark. Accordingly, we have rewritten how we additionally adjusted for the variables in 8 models more clearly (lines 206-211).
- the description of sensitivity analysis (section 3.6) is rather brief and would benefit from further detail including why the outcomes were limited and what the additional multivariate analysis included.
Response 2. We agree with the reviewer’s comment. Accordingly, we added further details about the sensitivity analysis in our study. We applied competing risk analyses based on Fine and Gray’s proportional subhazards model, because the competing risk of T2D and death may confound the estimated of risk for T2D as the outcome which means the death event that happened earlier to T2D event could prevent the individuals from T2D development. We showed that if we restricted the outcome to incident T2D and censored for death, the association between proinsulin and incident T2D did not change after further adjustments for all confounders similar to the primary analyses (lines 255-268).
- the authors acknowledge that generalising their data to other ethnicities is not possible given the predominantly white (99%) population studied. They do however, make reference to other studies in other ethnicities. If would be very useful for the reader to have a short description of the main findings of these studies.
Response 3. We thank the reviewer for this important suggestion. We added the main findings of previous studies which were performed in the population with different ethnicities (lines 357-361).
Reviewer 2 Report
Excelent manuscript ,
I suggest adding the following recent work to the bibliographical citations of the discussion:
The role of the α cell in the pathogenesis of diabetes: A world beyond the mirror
Martínez, M. S., Manzano, A., Olivar, L. C., Nava, M., Salazar, J., D’marco, L., Ortiz, R., Chacín, M., Guerrero-Wyss, M., Cabrera de Bravo, M., Cano, C., Bermúdez, V. & Angarita, L., 1 sept. 2021, En: International Journal of Molecular Sciences. 22, 17, 9504.
Author Response
Response to Reviewer 2 comment
Excelent manuscript ,
I suggest adding the following recent work to the bibliographical citations of the discussion:
The role of the α cell in the pathogenesis of diabetes: A world beyond the mirror
Martínez, M. S., Manzano, A., Olivar, L. C., Nava, M., Salazar, J., D’marco, L., Ortiz, R., Chacín, M., Guerrero-Wyss, M., Cabrera de Bravo, M., Cano, C., Bermúdez, V. & Angarita, L., 1 sept. 2021, En: International Journal of Molecular Sciences. 22, 17, 9504.
Response. We appreciate the kind words of the reviewer and thank the reviewer for the suggestion. To accommodate the comment of the reviewer, we have added the suggested article to the discussion of our manuscript (lines 292-299).